# Pulmonary Capacity, Blood Composition and Metabolism among Coal Mine Workers in High- and Low-Altitude Aboveground and Underground Workplaces

**DOI:** 10.3390/ijerph19148295

**Published:** 2022-07-07

**Authors:** Yi Wang, Hongchu Wang, Yinru Chen, Naxin Xu, Winson Lee, Wing-Kai Lam

**Affiliations:** 1Department of Physical Education, Renmin University of China, Beijing 100872, China; wang-yi@ruc.edu.cn; 2Sports and Social Development Research Center, Renmin University of China, Beijing 100086, China; 3School of Mathematical Sciences, South China Normal University, Guangzhou 510631, China; wang_hc@m.scnu.edu.cn; 4College of Education, Beijing Sport University, Beijing 100084, China; 2020210083@bsu.edu.cn; 5Sport Science School, Beijing Sport University, Beijing 100084, China; 2021210459@bsu.edu.cn; 6School of Mechanical, Materials, Mechatronic and Biomedical Engineering, University of Wollongong, Wollongong, NSW 2522, Australia; 7Sports Information and External Affairs Centre, Hong Kong Sports Institute, Hong Kong, China

**Keywords:** dust lung disease, occupation health, autonomic nervous system

## Abstract

(1) Background: While previous studies revealed how underground mining might adversely affect the cardiopulmonary functions of workers, this study further investigated the differences between under- and aboveground mining at both high and low altitudes, which has received little attention in the literature. (2) Methods: Seventy-one healthy male coal mine workers were recruited, who had worked at least 5 years at the mining sites located above the ground at high (>3900 m; *n* = 19) and low (<120 m; *n* = 16) altitudes as well as under the ground at high (*n* = 20) and low (*n* = 16) altitudes. Participants’ heart rates, pulmonary functions, total energy expenditure and metabolism were measured over a 5-consecutive-day session at health clinics. (3) Results: Combining the results for both above- and underground locations, workers at high-altitude mining sites had significantly higher peak heart rate (HR), minimum average HR and training impulse as well as energy expenditure due to all substances and due to fat than those at low-altitude sites. They also had significantly higher uric acid, total cholesterol, creatine kinase and N-osteocalcin in their blood samples than the workers at low-altitude mining sites. At underground worksites, the participants working at high-altitude had a significantly higher average respiratory rate than those at low-altitude regions. (4) Conclusion: In addition to underground mining, attention should be paid to high-altitude mining as working under a hypoxia condition at such altitude likely presents physiological challenges.

## 1. Introduction

Coal is an important source of energy, accounting for 27.62% of primary energy consumption worldwide [1]. That percentage has reached more than 60% in China [2,3]. There are over 140,000 coal mine workers across China and millions across the world [3,4]. These workers always work in extreme conditions at both aboveground and underground work sites.

Underground mining can present health risks to workers due to poor ventilation and dustiness [5,6]. These might be the reasons behind the high and increasing annual rates of coal-mining occupational health problems such as lung diseases (pneumoconiosis and silicosis), cardiopulmonary function injury and cardiovascular disease among mining workers [7,8,9,10]. In addition, underground miners usually work in high-humidity and high-temperature environments, which present additional risks to the cardiopulmonary functions [11].

Although ventilation can be better at aboveground mining sites, recent trends in developing high-altitude sites present another health challenges to workers. High altitude can lower oxygen saturation, which can cause acute mountain sickness, long-term chronic intermittent hypoxia, chronic intermittent hypobaric hypoxia, chronic mountain sickness (CMS) and high-altitude pulmonary hypertension. They can produce symptoms of cardiorespiratory capacity and metabolism [12,13,14]. Mining sites that are 2500 m above the sea level are considered to be high-altitude locations (West, 2002). While most previous studies looked into the health of coal-mining workers at lowߝmoderate altitude workplaces (<2500 m), little attention has been paid to high-altitude regions [14,15].

The effects of mining activities on cardiopulmonary functions have been investigated in previous studies, as reviewed in [16]. Cardiovascular fitness can be monitored by measuring the heart rate variability [17]. Meanwhile, cardiopulmonary performance can also be monitored by measuring oxygen uptake (VO_2_max) [18], which is widely used in sport science [19]. While previous studies investigated whether coal-mining activities decreased pulmonary function [20], these studies did not pay attention to any differences among mining worksites with high and low attitude as well as under and above the ground.

The energy expenditure of different coal-mining tasks has long been documented [21]. Other than supporting movements, the human body uses about 70% of energy (known as basal metabolism rate) to maintain vital functions such as breathing and keeping warm [22]. Knowing any differences in the basal metabolism rate of workers among various settings of the mining workplace could facilitate better guidelines on the nutritional requirements of mining workers. In addition, analysis of blood composition (red blood cell count and concentrations of hemoglobin, uric acid, creatine kinase and N-osteocalcin) can give a further insight regarding how the body responds to changes in a work environment in which oxygen and ventilation are compromised [23,24].

The objective of this study was to compare the heart rate variability, cardiorespiratory capacity and metabolism in coal mine workers at different altitudes (high vs. low altitude) and worksites (aboveground vs. underground). Based on the previous studies, it is expected that aboveground/high-altitude coal mine workers would demonstrate more red blood cells, higher energy expenditure, oxygen consumption and protein metabolism and smaller heart rates than the underground/low-altitude workers. The results from this study can provide insights into occupational health in coal mining.

## 2. Materials and Methods

### 2.1. Participants

A total of 71 healthy male coal mine workers, who were machine operators, were recruited from the coal mines located at Tibet (Huatailong Mining sites, altitude about 3990–4400 m) and Tangshan (Kailun Coal Mining site, altitude about 100–120 m). These two mining sites were selected as they passed the national safely accreditation. They also had the same mining equipment and were operated on the same working schedule. The participants were categorized into one of four groups, based on the altitude (high vs. low) and type of worksite (aboveground vs. underground) (Table 1): high altitude and aboveground (High-Above, *n* = 19), high altitude and underground (High-Under, *n* = 20), low and aboveground (Low-Above, *n* = 16) and low altitude and underground (Low-Under, *n* = 16). The participants in the High-Above and Low-Above groups were working only in aboveground mines, while the participants in High-Under and Low-Under were working in only underground mines during the entire employment period. There were no significant differences in height, bodyweight and body mass index (BMI) (*Ps* > 0.05) among the four groups. The participants were required to have at least five years of experience working and living close to the same mining work sites. They were excluded if they had changed their working environment from aboveground to underground or from underground to aboveground at any time. All participants had no coronary artery disease, anemia or other cardiovascular disease. They were free of any clinically evidenced history of chronic bronchitis. All participants provided their informed consent before any measurements in this study.

### 2.2. Procedure

All participants performed heart rate test, basal metabolism rate (BMR) test, pulmonary test, blood test and maximum oxygen consumption (VO_2_max) test over five consecutive days. They stayed overnight at the health clinics (high altitude (4250 m above the sea)—Health 100 Body Check Center, Huatailong Mining Company, Lasa, Tibet, China; low altitude (100 m above the sea)—KaiLuan General Hospital Body Check Center, Tangshan, Hebei, China). The participants were asked not to consume any alcohol or caffeine one day (after 4:00 p.m.) before the measurements. On the first day, we measured the height and bodyweight of all participants. At 8:00 p.m., the heartbeat monitor (Firstbeat Bodyguard 2, Firstbeat Technologies Oy, Finland) was attached to the participants to continuously monitor R-R interval heartbeat profiles over the coming 72 h (i.e., until day 4, 8:00 p.m.). The participants were asked to keep their phones and any other electrical devices with transmitters at least two meters away from the bed to avoid any electrical interfere with the recordings. To allow complete rest for the BMR test in the morning the following day, the participants lay down and minimized physical activity in low-illumination rooms (~22 °C; ~50% relative humidity) [22,25] and fasted after 8:00 p.m.

On day 2, in the morning, the Cortex Metalyzer (Cortex Metalyzer-3B, Cortex Biophysik, Leipzig, Germany) was used to measure the complete basal metabolism rate and pulmonary function with breath-by-breath gas-exchange monitoring simultaneously while the participants were fully awake. Before the measurement was initiated, participants were lying on the bed for 10 min to allow stable data collection. The participants were then instructed to adopt a supine position for data recording and remained at rest for 25 min [26]. A five-minute steady curve was automatically selected with the software of the Cortex Metalyzer (Cortex Metalyzer-3B, Cortex Biophysik, Leipzig, Germany) to obtain the BMR and pulmonary ventilation (VE) variables.

On day 3, in the morning, blood samples were obtained between 7:00 a.m. and 8:00 a.m. after an overnight fast. The samples were obtained from an antecubital vein and collected in a BD Vacutainer plasma tube (Becton, Dickinson and Company, Franklin Lakes, NJ, USA). Plasma was isolated by centrifugation at 3500 rpm for 10 min and frozen at −80 °C within two hours of collection [27]. Analyses were subsequently completed within three months of collection.

On day 4, the participants were instructed to avoid any intake of caffeine (8 h), medicine (12 h) and alcohol (24 h) and asked not to perform any vigorous physical activity for 12 h before the VO_2_max test on Day 5.

On day 5, the participants were instructed to perform a graded treadmill running test in accordance with the Bruce protocol to determine the maximal oxygen uptake (VO_2_max). The graded exercise test design was used to assess the cardiorespiratory fitness including VO_2_max and maximum aerobic speed (Vmax) [28]. At the beginning of the testing session, the participants performed a 5 min warm-up at 8 km h^−1^ on a motorized treadmill. The Cortex Metalyzer (Cortex Metalyzer-3B, Cortex Biophysik, Germany) was used for measurement in the breath-by-breath mode and the mean values were calculated from expired air at 30 s intervals [27].

### 2.3. Data Processing

The test variables were selected as they were related to the potential differences between underground and aboveground and between high-altitude and low-altitude regions [12,20,23,24,29,30]. For heart rate variability, a heart rate monitor (Firstbeat Bodyguard 2, Firstbeat Technologies Oy, Finland) was used to measure the heart rate (HR) over three consecutive days. The RMSSD (asleep/awake states), standard deviation of NN intervals (SDNN) and ratio of low- and high-frequency power bands (LF/HF) were calculated for different phases (i.e., diurnal phase and nocturnal phase) at 6:00–12:00 and 0:00–6:00, respectively [31,32]. Additionally, the training impulse (TRIMP) was determined in this study as it was defined as the training intensity multiplied over time [32]. We calculated the data from each 30 s epoch-by-epoch data, with >5% excluded for the analysis [32]. The corresponding durations of workout in each heart rate zone (>50% to 60%, >60% to 70%, >70% to 80%, >80% to 90% and >90%) were determined with the Firstbeat monitor. The TRIMP was the summation of all intensity levels greater than 50% maximum heart rate multiplied by the duration of workout in the respective heart rate zones according to the following formula [33]:$$\mathrm{TRIMP}=T\times [\frac{\left({\mathrm{HR}}_{\mathrm{ex}}-{\mathrm{HR}}_{\mathrm{rest}}\right)}{\left({\mathrm{HR}}_{\max}-{\mathrm{HR}}_{\mathrm{rest}}\right)}]\times 0.64e^{1.92[\frac{\left({\mathrm{HR}}_{\mathrm{ex}}-{\mathrm{HR}}_{\mathrm{rest}}\right)}{\left({\mathrm{HR}}_{\max}-{\mathrm{HR}}_{\mathrm{rest}}\right)}]}$$
where*T* = duration of the workout*HRex* = heart rate during workout*HRrest* = resting heart rate*HRmax* = maximal heart rate*e* = 2.718

For metabolism data, five-minute stable metabolism data from the Cortex Metalyzer were selected from the original data to calculate the basal metabolism rate (BMR). The BMR indicates the daily rate of energy metabolism an individual needs to sustain and preserve the integrity of vital functions [22]. In addition, the total energy expenditure due to all substances (protein, fat and carbohydrates) was devised and benchmarked by the Firstbeat company (Firstbeat Bodyguard 2, Firstbeat Technologies Oy, Jyväskylä, Finland), while the relative percentage of the amount of energy expenditure due to the fat and carbohydrates was estimated based on the level and duration of each heart rate zone [34]. For blood composition data, red blood cell count, hemoglobin concentration, uric acid, creatine kinase and N-osteocalcin were analyzed [23,24]. For pulmonary function data, average ventilation (VE) at rest/work, average respiratory rate, average VO_2_ and percent VO_2_max were analyzed [12,30].

### 2.4. Statistical Analyses

All statistical analyses were performed in the SPSS program (SPSS 20.0, SPSS Inc., Chiacago, IL, USA). The normality and homogeneity of variance were assessed with the Shapiro–Wilk test and Levene’s homogeneity test, respectively. Two-way Altitude (high vs. low) x Worksite (above vs. under) ANOVAs were performed to determine whether there is any significant interaction and main effects on all test variables. The main effects refer to any significant differences in the measured parameters (1) between the high and low altitudes and (2) between above and underground sites. The interaction looks into any significant differences among all 4 working conditions (High-Above, High-Under, Low-Above and Low-Under). When significant interaction was determined at *p* < 0.05, one-way ANOVA or Welch’s heteroscedastic F-test (if data normality/homogeneity of variance was violated) was then performed, followed by post hoc multiple comparisons (Bonferroni or Games–Howell test). The alpha was set at 0.05.

## 3. Results

### 3.1. Heart Rate (HR) Variables

The ANOVA indicated a significant interaction on average HR (*p* < 0.05, Table 2). Further analysis of the interaction revealed that at high-altitude regions, participants working aboveground (High-Above) had higher average HR than those at the underground worksites (High-Under, *p* < 0.05), while no such difference was found in the low-altitude region (*p* > 0.05). The significant main effects of altitude indicated that when compared with the low altitude (Low-Above and Low-Under), participants working at high altitude (High-Above and High-Under) displayed higher peak HR, minimum average HR, training impulse (TRIMP) and low power frequency to high power frequency (LF/HF) ratio (*p* < 0.05) but lower RMSSD-Sleep, RMSSD-Awake and SDNN values (*p* < 0.001).

The significant main effect of worksite revealed that underground participants (High-Under and Low-Under) had smaller peak HR, minimum average HR and TRIMP but larger RMSSD-Awake values than the aboveground participants (High-Above and Low-Above, *p* < 0.05, Table 2).

### 3.2. Metabolism Variables

The ANOVA indicated significant interaction on higher total energy expenditure due to EE-carbohydrates (*p* = 0.007, Table 3). Further analysis of the interaction revealed that at high-altitude regions, participants demonstrated higher EE-carbohydrates when working aboveground (High-Above) compared with the underground worksites (High-Under, *p* < 0.05), while no significant difference was found between underground and aboveground worksites at low-altitude regions (*p* > 0.05).

The main effect of altitude indicated the higher total energy expenditure due to all substances (EE-Total) and fat (EE-Fat) at high altitude (High-Above and High-Under) than in the low-altitude region (Low-Above and Low-Under, *p* < 0.05, Table 3). The main effect of worksite revealed the higher EE-Total and EE-Fat aboveground (High-Above and Low-Above) compared with the underground worksites (High-Under and Low-Under, *p* < 0.05, Table 3).

### 3.3. Blood Composition Variables

There were no significant interaction effects and main effects of worksite on any of the blood composition variables (*p* > 0.05, Table 4). The main effects of altitude indicated participants working at high altitude (High-Above and High-Under) demonstrated significantly higher uric acid, total cholesterol, creatine kinase and N-osteocalcin compared with those in the low-altitude region (Low-Above and Low-Under, *p* < 0.05, Table 4).

### 3.4. Pulmonary Function Variables

The ANOVA indicated a significant interaction on average respiratory rate (*p* = 0.008, Table 5). Further analysis of the interaction indicated that at the underground worksites, the participants had higher average respiratory rate when working at high altitude (High-Under) than in the low-altitude region (Low-Under, *p* < 0.05), but no differences between altitudes in aboveground worksites (*p* > 0.05).

The main effect of altitude indicated that the average pulmonary ventilation during rest and work and average oxygen consumption (VO_2_) in the high-altitude region (High-Above and High-Under) were higher compared with that in the low-altitude region (Low-Above and Low-Under, *p* < 0.05). The main effect of worksite indicated that participants had higher average pulmonary ventilation during work (i.e., average VE-work), average oxygen consumption and percentage of maximum oxygen consumption when working aboveground (High-Above and Low-Above) than in the underground worksites (High-Under and Low-Under, *p* < 0.05).

## 4. Discussion

Prolonged working at mining worksites (underground, aboveground, high or low altitudes) may impact health, which can be identified by some biomarkers such as heart rate variability, metabolism, blood composition and pulmonary capacity; comprehensive health care for workers requires deeper insights into the relationship among worksite, altitude and biological reaction. This study examined these biomarkers in coal mine workers who had at least five years of experience working in one of these worksites (high or low altitude, as well as above or underground). The current results supported the hypothesis that altitudes and types of worksites play some important roles in these markers.

Over half of the participants worked at high-altitude coal-mining sites (high altitude region, 4500 m above sea level), which create a condition of hypoxia. Although these workers have been working at the same worksites for more than 5 years, which may have already lowered the impact of hypoxia, these people still displayed significantly higher maximum average HR ratio, peak HR, minimum average HR, training impulse (TRIMP) and low power frequency to high power frequency (LF/HF) ratio, compared with their counterparts working at low altitudes. This suggested that having worked under the condition of high-attitude hypoxia for at least five years, such a condition still required the heart to work harder to provide the oxygen needs to sustain and preserve the integrity of vital functions [34,35]. This is in line with the mechanism proposed in previous studies, which indicated that the increased heart and respiratory rates under such conditions were due to the excitation of the peripheral chemoreceptors (e.g., carotid body) and central chemoreceptors, which then sent impulse along the sinus nerves and vagal nerves to the respiratory and cardiovascular systems [35,36]. It aligns with the practice of improving sport performance (top-level) at sea level by requiring athletes to live at high altitude to acclimate to the relative lack of oxygen for increasing the mass of red blood cells and hemoglobin as well as muscle metabolism and to train at low altitude [37]. Additionally, the higher TRIMP would suggest more physiological stimulus to trigger sympathetic excitation in the body at the synthetic stage. On the other hand, the coal mine workers working at high altitudes showed lower RMSSD-Sleep, RMSSD-Awake and SDNN values than those working at low altitudes. This could suggest poor heart resistance of these workers or more sympathetic-excitation under prolonged work in high-altitude regions. The heart rate parameters reveal the ability of the heart to respond to different contextual stimuli [31,32], which has been considered to be one of the key indicators to assess the general health of coal workers.

High-altitude workers also showed higher total energy expenditure due to all substances (EE-Total) and fat (EE-Fat) than those in the low-altitude regions. This could be related to cold temperature and lower oxygen concentration at high altitudes, which require more active fat metabolism for higher exergy expenditure needed to maintain the basic blood circulation, breathing, body temperature and internal body homeostasis [22,35]. Furthermore, the blood composition analysis revealed that participants working at high altitudes demonstrated significantly higher uric acid, total cholesterol, creatine kinase and N-osteocalcin than those in the low-altitude region. Higher blood uric acid and N-osteocalcin would be related to less air moisture. Other studies found that hematological responses at high altitudes (3800–4600 m) are gradual and characterized by increased hemoglobin and hematocrit concentrations over working time [24]. At sea level, arterial maximum exercise decreases oxygen saturation [29], which is caused by reduced diffusion capacity [38]. A similar condition appears in the workers under the condition of high-altitude hypoxia, which can lead to long-term changes in blood composition [39].

While most of the previous studies have investigated the changes in pulmonary function during maximum exercises [23], little research has been established on physical characteristics in the resting state. Our results indicated that healthy workers at high altitude had higher pulmonary ventilation during rest and work and average oxygen consumption than those in the low-altitude region. This would be associated with the lower oxygen saturation, which triggers greater demands on pulmonary function [20]. Previous studies reported that high altitude can lead to long-term chronic intermittent hypoxia, chronic intermittent hypobaric hypoxia and high-altitude pulmonary hypertension [12,13,14].

To our best knowledge, this is the first study to report the differences between aboveground and underground worksites in terms of the biomarkers of mining workers. The underground worker had smaller peak HR, minimum average HR and TRIMP and smaller total energy expenditure (EE-Total and EE-Fat) but larger RMSSD-Awake values than the aboveground participants. This indicates a lower activity intensity when working at underground worksites. The RMSSD reflects the tension of sympathetic and parasympathetic nerves, which are related to mental stress and psychological state [38]. From the psychological perspective, the higher RMSSD would be related to depression, irritability and psychological threats (accidence, darkness) in underground working environment [37,40,41]. Additionally, the underground workers had lower average pulmonary ventilation during work (i.e., average VE-work), average oxygen consumption and percentage of maximum oxygen consumption than those in the aboveground environment. The poor ventilation and dustiness might explain the lower pulmonary ventilation and working intensity in underground regions in underground mining sites [5,6,37].

Interestingly, there are some interaction effects between altitudes and worksites. Our study showed that at high-altitude regions, participants working aboveground had higher EE-carbohydrates than the underground region, while there was no significant difference between underground and aboveground at low-altitude regions. As high-altitude regions present thinner air and lower temperature, mining workers increased their breath rate and degree of anaerobic capacity. The superposition effect on physiology stimulus can be explained by the altitude and worksites. On the other hand, further analysis of the correlation of pulmonary ventilation levels and type of workplace revealed that, at the underground mining worksites, the participants had higher average respiratory rate when working in high altitude than in the low-altitude region, but no differences were found between altitude regions in aboveground worksites. This would suggest a limit to the breathing rate.

There are some limitations when interpreting our findings. First, only healthy coal miners were recruited, which makes this study not generalizable to all coal mine workers. Previous studies reported that the prolonged mining work may cause lung diseases (pneumoconiosis and silicosis), cardiopulmonary function injury and cardiovascular disease among mining workers [7,8,9,10]. Future epidemiological study should provide adequate insights to the field of industrial medicine, relating to occupational diseases such as coronary artery and lung diseases. Second, the actual amount of mining intensity was not controlled among different altitudes and worksites. Although the mining work content was standardized between the worksites in Tibet (altitude, 3990–4400 m) and Tangshan (sea level), the actual work intensity was based on the individual subjective intensity for their daily work. In the future, implementing computer vision technology would help to quantify the amount of work intensity of individual workers between worksites.

## 5. Conclusions

High-altitude mining workers can present higher HR, total energy expenditure, metabolism indicators and pulmonary function than those in the low-altitude region, which may be related to the hypoxic environment. Compared to the aboveground worksites, workers in underground mining demonstrated smaller HR, TRIMP and total energy expenditure. These findings provide some insights for designing the work schedule and intensity among workers at mining worksites.

## Figures and Tables

**Table 1 ijerph-19-08295-t001:** Demographic characteristics.

		High-Above (*n* = 19)	High-Under (*n* = 20)	Low-Above (*n* = 16)	Low-Under (*n* = 16)	*p*-Value
Age (year)		35.84 ± 5.5	36.7 ± 5.4	36.8 ± 5.8	36.7 ± 7.3	0.963
Height (cm)		173.7 ± 6.2	172.2 ± 4.7	171.0 ± 8.2	172.2 ± 5.4	0.625
Weight (kg)		73.6 ± 14.0	75.9 ± 12.3	70.7 ± 12.7	75.4 ± 11.2	0.628
BMI (kg/m^2^)		24.3 ± 3.9	25.6 ± 3.6	24.1 ± 3.3	25.3 ± 3.2	0.498
Smoking	no	5	5	4	6	-
(per day)	≤1 pack	13	15	9	13	-
	>1 pack	1	0	7	3	-

**Table 2 ijerph-19-08295-t002:** Heart rate (HR) variables.

		Altitude		*p*-Value		
	Worksite	High	Low	Interaction	Altitude	Worksite
Average HR (beat)	Under	85.85 ± 6.32	74.94 ± 4.33	**0.042**	<**0.001**	<**0.001**
	Above	96.21 ± 5.08	79.69 ± 6.62			
Peak HR (beat)	Under	132.00 ± 9.11	126.75 ± 6.15	0.078	<**0.001**	<**0.001**
	Above	153.68 ± 7.50	141.19 ± 10.56			
Minimum average HR (beat)	Under	59.75 ± 6.28	50.81 ± 2.83	0.285	<**0.001**	**0.012**
	Above	65.05 ± 8.94	53.00 ± 3.23			
TRIMP (Index)	Under	169.80 ± 160.26	45.29 ± 33.56	0.120	<**0.001**	<**0.001**
	Above	462.87 ± 219.33	206.67 ± 213.49			
RMSSD-Sleep (ms)	Under	22.01 ± 7.58	37.71 ± 14.56	0.553	<**0.001**	0.761
	Above	24.51 ± 10.97	36.90 ± 13.20			
RMSSD-Awake (ms)	Under	17.46 ± 7.22	23.16 ± 3.57	0.791	<**0.001**	**0.011**
	Above	14.19 ± 4.13	19.16 ± 7.16			
SDNN (ms)	Under	101.64 ± 15.89	145.73 ± 21.56	0.442	<0.001	0.696
	Above	99.57 ± 28.99	152.03 ± 22.49			
LF/HF ratio	Under	496.31 ± 253.09	295.30 ± 143.85	0.150	**0.003**	0.244
	Above	483.95 ± 168.84	411.06 ± 125.94			

Note: TRIMP = training impulse; RMSSD-Sleep and RMSSD-Awake = root mean square of successive difference during sleep and awake, respectively; SDNN = standard deviation of normal-to-normal heartbeat; LF/HF = low power frequency to high power frequency ratio. Bold numbers indicate significant differences/interactions.

**Table 3 ijerph-19-08295-t003:** Metabolism variables.

		Altitude		*p*-Value		
	Worksite	High	Low	Interaction	Altitude	Worksite
Basal metabolism rate (%)	Under	2041.25 ± 254.97	2147.21 ± 302.42	0.399	0.450	0.333
	Above	2032.90 ± 278.85	2027.06 ± 270.49			
EE-Total (kcal)	Under	3795.06 ± 606.38	3381.46 ± 794.10	0.706	**0.006**	**0.001**
	Above	4784.11 ± 1251.11	3968.13 ± 1202.84			
EE-Fat (kcal)	Under	2671.07 ± 494.27	2193.22 ± 652.26	0.400	**0.012**	**0.001**
	Above	3447.84 ± 1026.17	2822.67 ± 980.73			
EE-Carbohydrates (kcal)	Under	1123.99 ± 119.01	1188.24 ± 174.18	**0.007**	0.174	0.070
	Above	1336.27 ± 231.64	1145.46 ± 231.30			

Note: EE-Total, EE-Fat and EE-Carbohydrates = total energy expenditure due to utilization of all substances (protein, fat and carbohydrates), due to fat utilization and due to carbohydrate utilization, respectively. Bold numbers indicate significant differences/interactions.

**Table 4 ijerph-19-08295-t004:** Blood composition variables.

		Altitude		*p*-Value		
	Worksite	High	Low	Interaction	Altitude	Worksite
Red blood cell count (million/mm^3^)	Under	5.44 ± 0.68	5.28 ± 0.86	0.872	0.332	0.087
	Above	5.80 ± 0.88	5.58 ± 0.83			
Hemoglobin concentration (g/dL)	Under	155.30 ± 12.82	151.06 ± 19.60	0.627	0.468	0.581
	Above	155.53 ± 11.49	154.69 ± 14.05			
Uric acid (mg/dL)	Under	403.60 ± 64.11	364.81 ± 67.58	0.199	<**0.001**	0.538
	Above	431.53 ± 64.66	354.94 ± 43.45			
Total cholesterol (dL)	Under	4.64 ± 0.87	4.07 ± 0.86	0.826	**0.002**	0.672
	Above	4.76 ± 0.88	4.11 ± 0.54			
Creatine kinase (U/L)	Under	231.95 ± 159.23	133.25 ± 68.70	0.444	<**0.001**	0.975
	Above	254.89 ± 172.06	108.31 ± 58.91			
N-osteocalcin (mg/dL)	Under	27.55 ± 7.14	26.27 ± 6.73	0.056	**0.007**	0.637
	Above	29.80 ± 6.34	22.58 ± 5.03			

Note: EE-Total, EE-Fat and EE-Carbohydrates = total energy expenditure due to utilization of all substances (protein, fat and carbohydrates), due to fat utilization and due to carbohydrates utilization, respectively. Bold numbers indicate significant differences/interactions.

**Table 5 ijerph-19-08295-t005:** Pulmonary function variables.

		Altitude		*p*-Value		
	Worksite	High	Low	Interaction	Altitude	Worksite
Average VE-rest (million/mm^3^)	Under	11.32 ± 1.78	8.88 ± 0.92	0.984	<**0.001**	0.856
	Above	11.24 ± 1.55	8.81 ± 1.96			
Average respiratory rate (time/min)	Under	18.90 ± 2.22	16.13 ± 1.46	**0.008**	**0.003**	**0.025**
	Above	18.68 ± 2.21	18.50 ± 1.79			
Average VE-work (mg/dL)	Under	14.30 ± 3.21	10.69 ± 1.78	0.970	<**0.001**	**0.001**
	Above	16.79 ± 3.19	13.13 ± 3.01			
Average VO_2_ (mL/kg/min)	Under	7.41 ± 1.24	5.40 ± 0.59	0.521	<**0.001**	<**0.001**
	Above	8.99 ± 0.92	6.65 ± 1.31			
^ Percent VO_2_max (%)	Under	10.93 ± 1.76	10.45 ± 1.71	0.123	0.551	**0.001**
	Above	11.98 ± 1.74	13.06 ± 3.04			

Note: ^ represents the results based on Welch’s heteroscedastic *F*-test. Average VE-rest and VE-work = pulmonary ventilation during rest and work, respectively; VO_2_ = oxygen consumption; Percent VO_2_max = percentage of maximum oxygen consumption. Bold numbers indicate significant differences/interactions.

## Data Availability

The data presented in this study are available on request from the corresponding author. The data are not publicly due to privacy.

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
