# Peer review of "Pulmonary Capacity, Blood Composition and Metabolism among Coal Mine Workers in High- and Low-Altitude Aboveground and Underground Workplaces"

_ijerph, 2022, doi:10.3390/ijerph19148295_

Round 1

Reviewer 1 Report

It is well established that miners working high altitudes have higher HR and other pulmonary disorders. This study compare cardiopulmonary and other biological parameters among mine workers high and low altitude above-ground and under-ground, which has similar observations which others groups already presents while evaluating high altitude coal miners. 

The study is well organized and clearly  presented the data considering all statistical analysis. Study provides all inclusive and exclusive criteria used in selecting participants, however having few other important parameters should have included which might produce an biased results- 

1. Smoking history of participants

2. location of clinic (in terms of altitudes above ground) where participants tests were performed

3. Why only male participants were included?

4. Did participants continued working in mine while having test done for 5 days? If not, shifting participant to clinic which is at different altitude than min location might alter test results.

At many places results are presented without mentioning which group is compared to which group? For example - line 26-29 in Abstract-is this result from above ground? Please carefully go through the manuscript again to make sure results are clearly presented.

Author Response

Reviewer: 1

Comments to the Author

It is well established that miners working high altitudes have higher HR and other pulmonary disorders. This study compare cardiopulmonary and other biological parameters among mine workers high and low altitude above-ground and under-ground, which has similar observations which others groups already presents while evaluating high altitude coal miners. 

The study is well organized and clearly presented the data considering all statistical analysis. Study provides all inclusive and exclusive criteria used in selecting participants, however having few other important parameters should have included which might produce an biased results- 

 >> Thank you for your positive feedback on this paper. We have addressed each of the following comments accordingly.

  1. Smoking history of participants

 >> Thank you. The smoking history is now provided in Table 1 as you suggested.

  1. location of clinic (in terms of altitudes above ground) where participants tests were performed

 >> Thank you for clarifying this. We selected the clinic locations that are nearby the mining sites to prevent any significant changes in altitudes during the transportataion. The clinic at the high altitude (4,250 meters above sea level) is 3-minute travelling distance by shuttle bus to the Huatailong Mining site at Tibet. The clinic at the low altitude (about 100 meters above sea level) is the walking distance from the Kailuan Coal Mining site. Therefore, there were no attitude difference between the clinics and mining sites.

  1. Why only male participants were included?

 >> Thank you for asking this. We did not include the female participants because there were only a few female workers in both mining sites. In addition, these female miners are older and engaged only in lighter working intensity.

  1. Did participants continued working in mine while having test done for 5 days? If not, shifting participant to clinic which is at different altitude than min location might alter test results.

 >> Thank you for clarifying this. All the participants lived in the staff apartments provided by the mining company which were nearby the clinics (same altitude between the clinics and mining site). To minimize the influence to their regular working intensity and allow monitoring the heart rate variability for 72 hours, the participants continued to work in the mining site during the days when the test was done (except for Day 4 and 5). The working arrangement is explained as the follows:

Day 1 – Arrive our clinics after regular work and stay over night

Day 2 – Complete basal metabolism rate and pulmonary function at the morning and then go to regular work

Day 3 – Collect blood samples at the morning and then go to regular work

Day 4 – Day-off (resting day) to avoid vigorous physical activity for the VO2max test at Day 5

Day 5 – Complete graded treadmill running test to measure VO2max at the morning and take the day-off

At many places results are presented without mentioning which group is compared to which group? For example - line 26-29 in Abstract-is this result from above ground? Please carefully go through the manuscript again to make sure results are clearly presented.

 >> Thank you for pointing this out. We have specified the group information (High-Above, High-Under, Low-Above, and Low-Under) in Abstract, Results and other sections where appropriate. For the main effect results (Altitude effect or Worksite effect), we have now provided the group information such as High-Above and High-Under for high altitude workers.

Reviewer: 1

Comments to the Author

It is well established that miners working high altitudes have higher HR and other pulmonary disorders. This study compare cardiopulmonary and other biological parameters among mine workers high and low altitude above-ground and under-ground, which has similar observations which others groups already presents while evaluating high altitude coal miners. 

The study is well organized and clearly presented the data considering all statistical analysis. Study provides all inclusive and exclusive criteria used in selecting participants, however having few other important parameters should have included which might produce an biased results- 

 >> Thank you for your positive feedback on this paper. We have addressed each of the following comments accordingly.

  1. Smoking history of participants

 >> Thank you. The smoking history is now provided in Table 1 as you suggested.

  1. location of clinic (in terms of altitudes above ground) where participants tests were performed

 >> Thank you for clarifying this. We selected the clinic locations that are nearby the mining sites to prevent any significant changes in altitudes during the transportataion. The clinic at the high altitude (4,250 meters above sea level) is 3-minute travelling distance by shuttle bus to the Huatailong Mining site at Tibet. The clinic at the low altitude (about 100 meters above sea level) is the walking distance from the Kailuan Coal Mining site. Therefore, there were no attitude difference between the clinics and mining sites.

  1. Why only male participants were included?

 >> Thank you for asking this. We did not include the female participants because there were only a few female workers in both mining sites. In addition, these female miners are older and engaged only in lighter working intensity.

  1. Did participants continued working in mine while having test done for 5 days? If not, shifting participant to clinic which is at different altitude than min location might alter test results.

 >> Thank you for clarifying this. All the participants lived in the staff apartments provided by the mining company which were nearby the clinics (same altitude between the clinics and mining site). To minimize the influence to their regular working intensity and allow monitoring the heart rate variability for 72 hours, the participants continued to work in the mining site during the days when the test was done (except for Day 4 and 5). The working arrangement is explained as the follows:

Day 1 – Arrive our clinics after regular work and stay over night

Day 2 – Complete basal metabolism rate and pulmonary function at the morning and then go to regular work

Day 3 – Collect blood samples at the morning and then go to regular work

Day 4 – Day-off (resting day) to avoid vigorous physical activity for the VO2max test at Day 5

Day 5 – Complete graded treadmill running test to measure VO2max at the morning and take the day-off

At many places results are presented without mentioning which group is compared to which group? For example - line 26-29 in Abstract-is this result from above ground? Please carefully go through the manuscript again to make sure results are clearly presented.

 >> Thank you for pointing this out. We have specified the group information (High-Above, High-Under, Low-Above, and Low-Under) in Abstract, Results and other sections where appropriate. For the main effect results (Altitude effect or Worksite effect), we have now provided the group information such as High-Above and High-Under for high altitude workers.

Reviewer 2 Report

his is a descriptive study with a small sample of healthy mining workers who are measured for a range of physiological and analytical parameters.

More specifically, I would like to point out that the sample is small, the results of nonparametric tests have much less value than in large samples, so all the results, although methodologically correct, cannot be extrapolated to a more general context. The tables are adequate and the data are well presented. That is why I considered in my previous report that it was appropriate for publication, but the paper does not give much more...

Author Response

Reviewer: 2

Comments to the Author

This is a descriptive study with a small sample of healthy mining workers who are measured for a range of physiological and analytical parameters.

More specifically, I would like to point out that the sample is small, the results of nonparametric tests have much less value than in large samples, so all the results, although methodologically correct, cannot be extrapolated to a more general context. The tables are adequate and the data are well presented. That is why I considered in my previous report that it was appropriate for publication, but the paper does not give much more...

>> Thank you for your time and effort on this manuscript and your recommendation for “appropriate for publication”. We also agree with you that dataset with larger samples will be of greater values of the small samples. After this preliminary work presented in this manuscript, we are planning to secure more funding to carry out coal-mining research with larger sample sizes in the near future.

Reviewer 3 Report

Thank you very much for the opportunity to review the manuscript "Pulmonary Capacity, Blood Composition and Metabolism among Coal Mine Workers in High and Low Altitude Above-ground and Under-ground Workplaces" by Y. Wang et al.

The authors compare several cardiovascular and metabolism parameters in coal mine workers at high and low altitude and in above and underground mining sites.

The presented results indicate significant differences in heart rate parameters, energy expenditure and metabolism parameters between participants from high and low altitude working sites and also between above ground and underground sites. Blood composition variables were found to be associated with altitude levels only.

The authors conclude that high altitude induces physiological effects that may be related to hypoxia. A lower observed heart rate, training impulse and energy expenditure was observed in underground miners and the authors see this as in possible indication that lower working intensity can be achieved in underground mining. The authors further propose their study as a scientific base for designing working schedules at coal mines.

The English writing is quite good, but there are several grammar/wording related points that have been outlined below.

This study is comparing at the state of cardiovascular/metabolic fitness in workers at the four conditions/sites, each represented by less than 21 healthy participants. Relatively low numbers of participants in combination with the possibility of non-random participant selection method limit the representative properties of this analysis.

The authors use ANOVA to statistically investigate differences in health parameters at given conditions. But also "interaction" is provided with a separate p-value. How was "interaction" calculated? ANOVA (analysis of variance) is testing differences, typically between multiple mean values. The term "interaction, indicated by ANOVA", is used in line 181 and 229 again. Please explain/correct.

line 162: "energy expenditure due to fat and carbohydrates were devised and benchmarked by the Firstbeat company". As we understand, this sentence is not correct. Energy expenditure was, in our understanding, estimated using the heart rate data obtained by the Firstbeat Bodyguard 2 system. The Firstbeat whitepaper for this estimation procedure does not mention any differentiation for fat or carbohydrates. Anyway, the performed estimation/calculation should be included in the material and methods section in a well comprehensible manner.

There are several points relevant to the comparability of the conditions of interest that have not been addressed by the authors:

1. Mining workers at high altitudes typically have different working schedules (i.e. longer recreation periods), due to the fact that the working performance is limited under these conditions. Is this also the case here? How do the working schedules compare to low altitudes?

2. "Mining worker" is generalizing term. Typically, there are very different tasks that are in the hands of specialized workers with very different level of environmental exposure and physical exertion. Results may thus be highly misleading when comparing, for example, coal hewers with mine-truck drivers.

3. Underground coal mines are typically in danger of eruptions from methane (firedamp) and also other gases. Often, oxygen is directly pumped/released into mines to guarantee the safety of underground miners. In the given context, it is important to know if this has been checked/considered in this investigation.

4. It is also important to know, if the mining workers at high altitude are commuting to their workplace from lower areas or if they constantly live at these elevated levels.

As being tolerable to hypoxia may be a prerequisite for working five years in high altitude, hypoxia in underground mines may show less impact in these pre-selected individuals. Such implications should also be discussed.

Line 320: We cannot comprehend the suggestion that the "live high/train low" concept is supported by the fact that underground workers have a higher respiratory rate at higher altitudes. As a higher higher breathing rate is commonly observed at higher altitude, it is expected to be higher also underground (and even more so as we expect additional hypoxia there). It is rather very surprising and contradictory to existing data that it has not been observed in above-ground mines.

Line 59,247/248: Heart rate variability can be an indicator of cardiovascular fitness (as well pointed out by H.C.D. Souza in lit.18). However, heart rate variability shows the ability of the heart to adapt to physiological challenge/activity) in terms of changes in heart rate. It should therefore be reflecting the difference between two states of activity and is thus not reflected by an average measurement heart rate, as shown in table 2. Heart rate variability is typically shown by (delta, range of) values in milliseconds and shows high individual variability as well as fluctuation over time. It therefore requires rather large sample numbers for statistical evaluation, that are not provided in this study..

"Additionally, training impulse (TRIMP) was determined in this study as it defined as training intensity multiplied over time". As we understand, what the authors did is multiplying heart rate with time. We understand this represents TRIMP(avg). The activity/monitoring phases are however not reflected/shown in the TRIMP-Index. Does the given value include both, recreational and working phases? The working/recreation circle for the observation period (72h) should be shown.

The observed physiological effects of high altitude are well known.

The conclusion, that lower working intensity can be achieved in underground mining, based on these non-representative results is rather doubtful.

Minor points:

line 28: "...due to all substances and fat..."

line 118 : "...fastafter.."

line 314: "..degree of anaerobic.."

line 321: ."..suggested that people live at high altitude to acclimate to the relative lack of oxygen for increasing the mass of red blood cells and haemoglobin as well as muscle metabolism, and train at low altitude, usually with the goal of improving performance at sea level." Better "For improvement of (top-level) sport performance at sea level, it has been suggested to..."

line 316: "..interaction on average respiratory.." Better: "further analysis of the correlation of pulmonary ventilation levels and type of workplace revealed that.."

The authors may consider discussing/citing the paper: Biol Res 46: 59-67, 2013
Acclimatization to chronic intermittent hypoxia in mine workers: a
challenge to mountain medicine in Chile
Jorge G. Farías, Daniel Jimenez, Jorge Osorio, Andrea B. Zepeda, Carolina A. Figueroa1 and Victor M. Pulgar

Author Response

Reviewer: 3

Comments to the Author

Thank you very much for the opportunity to review the manuscript "Pulmonary Capacity, Blood Composition and Metabolism among Coal Mine Workers in High and Low Altitude Above-ground and Under-ground Workplaces" by Y. Wang et al. The authors compare several cardiovascular and metabolism parameters in coal mine workers at high and low altitude and in above and underground mining sites. The presented results indicate significant differences in heart rate parameters, energy expenditure and metabolism parameters between participants from high and low altitude working sites and also between above ground and underground sites. Blood composition variables were found to be associated with altitude levels only.

The authors conclude that high altitude induces physiological effects that may be related to hypoxia. A lower observed heart rate, training impulse and energy expenditure was observed in underground miners and the authors see this as in possible indication that lower working intensity can be achieved in underground mining. The authors further propose their study as a scientific base for designing working schedules at coal mines.

 >> Thank you for your review, summary and comments on our work. We have addressed your comments as below.

The English writing is quite good, but there are several grammar/wording related points that have been outlined below.

 >> Thank you for the specific comments on the grammars. We have revised accordingly.

This study is comparing at the state of cardiovascular/metabolic fitness in workers at the four conditions/sites, each represented by less than 21 healthy participants. Relatively low numbers of participants in combination with the possibility of non-random participant selection method limit the representative properties of this analysis.

 >> Thank you for the comments. We also agree that more participants would increase the likelihood of yielding more useful information. Meanwhile, our sample size (a total of 71 participants in 4 groups) was comparable to previous studies (n=56, Bobo et al., 1983; n=73, Hong et al., 2019; n=16, Moraga et al., 2017; n=29, Richalet et al., 2002). While this study gave new insight into the health-related status of workers in very different mining work conditions, there were difficulties in recruitment given the 5-day commitments to this study and logistic arrangements in managing the measurements in underground and aboveground mining sites in both high and low altitudes.

The studies with comparable sample size:

Bobo M, Bethea NJ, Ayoub MM, Intaranont K. (1983). Energy Expenditure and Aerobic Fitness of Male Low Seam Coal Miners. Human Factors. 25(1),43-48. https://doi:10.1177/001872088302500104

Hong SH, Yang HI, Kim D, Gonzales TI, Brage S, Jeon JY. (2019). Validation of submaximal step tests and the 6-min walk test for predicting maximal oxygen consumption in young and healthy participants. International Journal of Environmental Research and Public Health. 16, 4858. https://doi:10.3390/ijerph16234858.

Moraga F.A., Osorio J., Calderón-Jofré R. & Pedreros A. (2018). Hemoconcentration during maximum exercise in miners with chronic intermittent exposure to hypobaric hypoxia (3800 m). High Altitude Medicine & Biology, 19(1), 15-20. https://doi.org/10.1089/ham.2017.0011.

Richalet J-P, Donoso MV, Jimenez D, Antezana A-M, Hudson C, Cortes G, Osorio J, Leon A. (2002). Chilean miners commuting from sea level to 4500m: A prospective study. High Altitude Medicine & Biology. 3(2), 159-166.

The authors use ANOVA to statistically investigate differences in health parameters at given conditions. But also "interaction" is provided with a separate p-value. How was "interaction" calculated? ANOVA (analysis of variance) is testing differences, typically between multiple mean values. The term "interaction, indicated by ANOVA", is used in line 181 and 229 again. Please explain/correct.

 >> Thank you for clarifying this. Two-way Altitude (high vs. low) x Worksite (above vs. under) ANOVAs were performed to determine if there is any significant interaction and main effects on all test variables. The main effects refer to any significant differences in the measured parameters 1) between the high and low altitudes and 2) between above and under- grounds. The interaction looks into any significant differences among all 4 working conditions (High-Above, High-Under, Low-Above and Low-Under).

Specifically, in Two-way ANOVA the total variation (total sum of square, SST) of the data was decomposed into the sum of squares (SSA) of factor A (Altitude), the sum of squares of factor B (Worksite) (SSB), the sum of squares of interactions (SSA+B) and the sum of squared residuals (SSE). The sum of squared interactions is calculated from the mean of each variable (one treated variable is a combination of factors A and B) subtracting the mean value of the factor A and mean value of the factor B and then adding the total mean and total sum of squares. The interaction test in the two-way ANOVA constructs the F statistic from the sum of squares of the interaction. When the interaction effect is significant (P-value<α), it means that the effect of factor A will have different effects on indicators due to the existence of factor B, or the factor B will have different effects on indicators because of the presence of factor A. When the interaction is not significant, it means that factor A and factor B may have only main effects.

line 162: "energy expenditure due to fat and carbohydrates were devised and benchmarked by the Firstbeat company". As we understand, this sentence is not correct. Energy expenditure was, in our understanding, estimated using the heart rate data obtained by the Firstbeat Bodyguard 2 system. The Firstbeat whitepaper for this estimation procedure does not mention any differentiation for fat or carbohydrates. Anyway, the performed estimation/calculation should be included in the material and methods section in a well comprehensible manner.

 >> Thank you for the suggestion. In this study, we estimated energy expenditure using the heart rate data according to the Firstbeat whitepaper (Energy Expenditure Estimation Method Based on Heart Rate Measurement Published: Feb 2007. Last update: March 2012). The estimation procedure determines total MET value or maximum oxygen uptake through heart rate and heart rate variability.

Furthermore, the relative percentage of the amount of fat and carbohydrates can be estimated according to the different ratios of glucose and lipid metabolism under different VO2 max percentages (or METs), respiratory quotient and caloric equivalent (McArdle et al., 2008).  The corresponding statement now read:

            “the total energy expenditure due to all substances (protein, fat and carbohydrates) was devised       and benchmarked by the Firstbeat company (Firstbeat Bodyguard 2, Firstbeat Technologies Oy,        Finland), while the relative percentage of the amount of energy expenditure due to the fat and             carbohydrates were estimated based on the level and duration of each heart rate zone.” (2nd         paragraph of Data Processing section)

The additional textbook and the figures below are now provided to describe the estimation.

  • McArdle WD, Katch FI, Katch VL (2008). Sports and Exercise Nutrition. LWW, Philadelphia, United States.

There are several points relevant to the comparability of the conditions of interest that have not been addressed by the authors:

  1. Mining workers at high altitudes typically have different working schedules (i.e. longer recreation periods), due to the fact that the working performance is limited under these conditions. Is this also the case here? How do the working schedules compare to low altitudes?

 >> Thank you for clarifying this. The two studied mining sites (Tibet and Tangshan) are owned and operated by the Chinese Government. All the mine workers were eating, living and working together within the same community (within the mining sites). All the mining equipment and the working schedules (such as working hour, lunch time, holiday) are identical across all mining sites (Tibet and Tangshan) operated by the Government.

The corresponding statement now provide: “These two mining sites were selected as they passed the national safely accreditation. They also had the same mining equipment and were operated at the same working schedule.”. (Participant section)

  1. "Mining worker" is generalizing term. Typically, there are very different tasks that are in the hands of specialized workers with very different level of environmental exposure and physical exertion. Results may thus be highly misleading when comparing, for example, coal hewers with mine-truck drivers.

 >> Thank you for the comments. We agree with you about the differences in manual and machine operated tasks.  The mining sites in China are highly machine-based. As such, all participants recruited were machine operators and frontline staff. It is now indicated in the 2.1 Participants section.

The corresponding statements now read:

  • “A total of 71 healthy male coal mine workers, who were machine operators, were recruited from the coal mines located at Tibet (Huatailong Mining sites, altitude about 3990-4400m) and Tangshan (Kailun Coal Mining site, altitude about 100-120m).

  1. Underground coal mines are typically in danger of eruptions from methane (firedamp) and also other gases. Often, oxygen is directly pumped/released into mines to guarantee the safety of underground miners. In the given context, it is important to know if this has been checked/considered in this investigation.

 >> Thank you for asking this. Yes, we checked this information before the start of the study. All state-owned mining sites in China have designed and operated with the strict standard ventilation facilities and environmental controls included oxygen concentration and methane detection. The two mining sites studied have passed the accreditation reviewed by a third party China University of Mining and Technology to ensure the national security standards. The corresponding statements now add:

            “These two mining sites were selected as they passed the national safely accreditation and had      the same mining equipment and were operated with the same working schedule.” (Participant   section)

  1. It is also important to know, if the mining workers at high altitude are commuting to their workplace from lower areas or if they constantly live at these elevated levels.

As being tolerable to hypoxia may be a prerequisite for working five years in high altitude, hypoxia in underground mines may show less impact in these pre-selected individuals. Such implications should also be discussed.

 >> Thank you for the comments. As staff apartments were provided to all coal mine workers in the mining sites, our participants have been working and living close to the same mine sites since they had joined the mining company.  All the participants reported no relocations from high altitude to low altitude region (High-Above and High-Under groups) or from low altitude to high altitude (Low-Above and Low-Under groups).

The corresponding implications are the prerequisite for working over 5 years in high altitude are now added at the 2nd last paragraph of the Discussion as you suggested:

            “Although these workers have been working in the same worksites for more than 5 years which may have already lowered the impact of hypoxia, these people still….

Line 320: We cannot comprehend the suggestion that the "live high/train low" concept is supported by the fact that underground workers have a higher respiratory rate at higher altitudes. As a higher higher breathing rate is commonly observed at higher altitude, it is expected to be higher also underground (and even more so as we expect additional hypoxia there). It is rather very surprising and contradictory to existing data that it has not been observed in above-ground mines.

 >> Thank you for the suggestion. The contradicting results would be due to the differences in environment conditions between underground and above-ground (ventilation and temperature) and the participants’ health status among studies. The air composition, temperature and humidity were well controlled underground while they cannot be controlled in open area (above-ground), resulting in respective physiological adaptations. While a lot of the previous mine-worker studies investigated workers with certain diseases, our study included only the healthy workers who had no coronary artery disease, anaemia or other cardiovascular disease.

The following statement was moved to the 2nd paragraph of the Discussion section when interpreting the main altitude effects:

             “It aligns with practice in improving (top-level) sport performance at sea level to require people to live at high altitude to acclimate to the relative lack of oxygen for increasing the mass of red blood cells and haemoglobin as well as muscle metabolism, and train at low altitude [43].”

Line 59,247/248: Heart rate variability can be an indicator of cardiovascular fitness (as well pointed out by H.C.D. Souza in lit.18). However, heart rate variability shows the ability of the heart to adapt to physiological challenge/activity) in terms of changes in heart rate. It should therefore be reflecting the difference between two states of activity and is thus not reflected by an average measurement heart rate, as shown in table 2. Heart rate variability is typically shown by (delta, range of) values in milliseconds and shows high individual variability as well as fluctuation over time. It therefore requires rather large sample numbers for statistical evaluation, that are not provided in this study..

 >> Thank you for asking this. While averaged heart rate is normally used to determine energy consumption, heart rate variability measured in our manuscript was used to reflect the activities of the autonomic nervous system which regulates the heart rate, blood pressure and breathing. In our study, we attached the firstbeat monitor to each of the participants at the Day 1 night and measure the heart rate over a consecutive 72-hour period.

"Additionally, training impulse (TRIMP) was determined in this study as it defined as training intensity multiplied over time". As we understand, what the authors did is multiplying heart rate with time. We understand this represents TRIMP(avg). The activity/monitoring phases are however not reflected/shown in the TRIMP-Index. Does the given value include both, recreational and working phases? The working/recreation circle for the observation period (72h) should be shown.

The observed physiological effects of high altitude are well known.

 >> Thank you for the comments. The heart rate was monitored for a consecutive 72-hour period (including normal working and leisure activities as detailed in the above response) for all participants.  The TRIMP reported in Table 2 refers to the total training impulses for all activities greater than 50% maximum heart rate during the 72-hour period.

After the duration of workouts for respective heart rate zones can be determined with the Firstbeat monitor, the total TRIMP was calculated as the summation of the training impulse at each heart rate zone as follows:

TRIMP1 = Heart rate Zone 1 (>50 to 60%) x Duration of workout 1

+ TRIMP2 = Heart rate Zone 2 (>60 to 70%) x Duration of workout 2

+ TRIMP3 = Heart rate Zone 3 (>70 to 80%) x Duration of workout 3

+ TRIMP4 = Heart rate Zone 4 (>80 to 90%) x Duration of workout 4

+ TRIMP5 = Heart rate Zone 5 (>50 to 60%) x Duration of workout 5

The corresponding statements are now added:

“The corresponding duration of workout in respective heart rate zones (>50 to 60%, >60 to 70%, >70 to 80%, >80 to 90% and >90%) were determined with the Firstbeat monitor. The TRIMP was the summation of all intensity levels greater than 50% maximum heart rate multiplied by the duration of workout in respective heart rate zones with the following formula…” (1st paragraph of 2.3 Data Processing section)

The conclusion, that lower working intensity can be achieved in underground mining, based on these non-representative results is rather doubtful.

 >> Thank you for the comments. The corresponding statement is now removed from the Conclusion.

Minor points:

line 28: "...due to all substances and fat..."

 >> Thank you for the comments. The corresponding phase is now revised as “due to all substances and due to fat”.

line 118 : "...fastafter.."

 >> The typo is now corrected

line 314: "..degree of anaerobic.."

 >> Thank you. The term is now revised as “degree of anaerobic capacity” to improve clarify.

line 321: ."..suggested that people live at high altitude to acclimate to the relative lack of oxygen for increasing the mass of red blood cells and haemoglobin as well as muscle metabolism, and train at low altitude, usually with the goal of improving performance at sea level." Better "For improvement of (top-level) sport performance at sea level, it has been suggested to..."

 >> Thank you. It is revised as you have suggested.

line 316: "..interaction on average respiratory.." Better: "further analysis of the correlation of pulmonary ventilation levels and type of workplace revealed that.."

 >> Thank you. It is revised as you have suggested.

The authors may consider discussing/citing the paper: Biol Res 46: 59-67, 2013
Acclimatization to chronic intermittent hypoxia in mine workers: a
challenge to mountain medicine in Chile
Jorge G. Farías, Daniel Jimenez, Jorge Osorio, Andrea B. Zepeda, Carolina A. Figueroa1 and Victor M. Pulgar

 >> Thank you. The review paper is now cited.

Round 2

Reviewer 3 Report

The authors have significantly improved the manuscript, mainly by adding and clarifying information on applied methods like TRIMP calculation. Still not all questions have been answered regarding the participants living and working situations, but I agree that these missing parts are not of such importance that should prevent publication. However, we still noticed some problems in language and  layout (f.e. Table 1  heading) that may require some editorial attention.